# TSTTC: A Large-Scale Dataset for Time-to-Contact Estimation in Driving Scenarios

**Yuheng Shi**
Tianjin University
yuheng@tju.edu.cn

**Zehao Huang**
TuSimple
zehaohuang18@gmail.com

**Yan Yan**
TuSimple
yanyan.paper@outlook.com

**Naiyan Wang**
TuSimple
winsty@gmail.com

**Xiaojie Guo**
Tianjin University
xj.max.guo@gmail.com

## Abstract

Time-to-Contact (TTC) estimation is a critical task for assessing collision risk and is widely used in various driver assistance and autonomous driving systems. The past few decades have witnessed development of related theories and algorithms. The prevalent learning-based methods call for a large-scale TTC dataset in real-world scenarios. In this work, we present a large-scale object oriented TTC dataset in the driving scene for promoting the TTC estimation by a monocular camera. To collect valuable samples and make data with different TTC values relatively balanced, we go through thousands of hours of driving data and select over 200K sequences with a preset data distribution. To augment the quantity of small TTC cases, we also generate clips using the latest Neural rendering methods. Additionally, we provide several simple yet effective TTC estimation baselines and evaluate them extensively on the proposed dataset to demonstrate their effectiveness. The proposed dataset and code are publicly available at dataset link and code link.

## 1 Introduction

In recent years, there has been a growing trend towards equipping vehicles with Advanced Driver Assistance System (ADAS), which consists of several subsystems such as Adaptive Cruise Control (ACC), Automated Emergency Braking (AEB) and Forward Collision Warning (FCW). ADAS aims to detect potential hazards as quickly as possible and alert the driver, or take corrective action to improve driving safety. AEB and FCW are critical features of ADAS that protect drivers and passengers and prevent traffic accidents. They both rely on the estimation of Time-to-Contact (TTC) which is defined as the time for an object to collide with the observer's plane. Although TTC can be predicted using data from various sensors, such as LiDAR, radar or camera. Vision-based methods are particularly attractive due to their low-cost and have been a popular choice among ADAS designers and manufacturers. Even in high-level (L3+) autonomous driving system, direct TTC estimation could also serve as an redundant observation when other distance measuring sensors fail.

Prior to the widespread adoption of deep learning, numerous vision-based theories and algorithms [25, 29, 10, 6, 4] for estimating TTC had been proposed. These algorithms are not data-driven, and usually rely on hand-crafted cues. Recently, several deep learning based TTC estimation algorithms have emerged [1, 45] and demonstrate promising results in driving scenarios. The emergence of deep

Submitted to the 38th Conference on Neural Information Processing Systems (NeurIPS 2024) Track on Datasets and Benchmarks. Do not distribute.

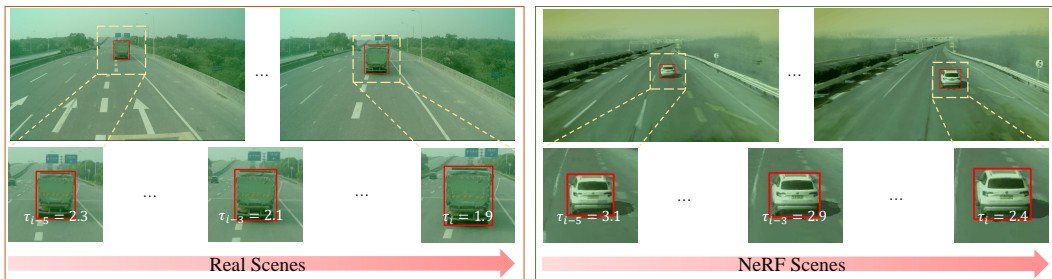

Figure 1: Example sequences and annotations from our dataset. The $\tau$ denotes the TTC ground-truth while the subscript denotes the frame index. We could observe that, the scale of the object increases as the TTC decreases.

learning has brought more powerful tools to computer vision and also brought higher demands for large-scale datasets. However, due to the lack of large-scale TTC datasets that capture real-world driving scenarios, these methods have to pre-train their model on synthetic flow datasets [27, 12, 22].

In this paper, we primarily address the challenge of TTC estimation in highway scenarios. Compared to urban scenarios, high speed driving in highway exhibits longer braking distances, thereby necessitating a broader range of perception capabilities. In order to facilitate the development of vision-based TTC estimation algorithms, we propose a large-scale monocular TTC dataset in this paper, using a class 8 heavy truck as the data collection platform. From raw data collected in urban and highway scenarios, we identify over 200K sequences covering a depth range of 400 meters. Each sequence contains six consecutive frames captured at a rate of 10Hz, with 2D, 3D bounding box and TTC ground-truth provided for a single object in each frame. Additionally, to address the limited availability of samples in rare scenarios, such as sudden braking (e.g. small TTC cases), we utilize Neural Radiance Fields (NeRF) [30] to generate additional data. These artificially generated data can be seamlessly integrated into our dataset, thereby increasing the quantity and diversity of data available for training. Fig.1 illustrates two typical examples from our dataset: vehicles are gradually approaching in real scenes and NeRF scenes, respectively. Besides the proposed dataset, we focus on object level TTC estimation rather than pixel level TTC estimation in [1]. Specifically, we provide a sequence of images for a certain object and ask for estimating the TTC value of it in the last frame. 2D Bounding boxes are available as optional inputs. Then we provide simple yet effective baselines based on the relationship between the scale ratio of objects in adjacent frames and TTC. We reformulate the problem as choosing the scale with the highest similarity in adjacent frames. Inspired by recent studies [1, 2], we further transform scale estimation from a regression problem into a set of binary classification tasks. A series of quantitative experiments are conducted to demonstrate the effectiveness and feasibility of our proposed techniques. Our main contribution can be summarized as follows:

- We propose a large-scale monocular TTC dataset for driving scenarios and will make it publicly available along with relevant toolkits to facilitate the development of TTC estimation algorithms for driving scenes.

- We propose two simple yet effective TTC estimation algorithms and extensively test them on the dataset to validate their effectiveness, which could serve as baseline methods for future study.

## 2 Related work

**Task and Methods.** In the scheme of monocular TTC estimation, TTC describes the time that an object will cross the camera plane under concurrent relative velocity. Denote the depth of an object in the camera coordinate as $y$, the time for the object under the current velocity to cross the camera plane could be calculated by:

$$\tau = -y/\frac{dy}{dt} = -y/\dot{y}, \tag{1}$$

where $\dot{y}$ is the relative velocity between the object and the camera. Though estimating either velocity or depth is an ill-posed problem, TTC can be estimated from images directly because it only depends on the ratio of them. Researchers have proposed various approaches to accomplish TTC estimation.

A viable approach is to utilize hand-crafted features such as closed contours, optical flow, brightness, or intensity from images [10, 8, 19, 29, 37, 43, 20]. Mobileye [10] adopted geometric information of the vehicles in image to estimate TTC by establishing the relationship between TTC and the width of vehicle. In addition to geometric-based methods, several studies have been proposed to address the task of TTC estimation using photometric-based features, without relying on geometric features or high-level processing. For instance, [19] adopted accumulated sums of suitable products of image brightness derivatives from time varying images to determine the TTC value. Furthermore, [43] elucidates the relationship between TTC and the changes in intensity in images. However, these hand-crafted features need carefully tuned parameters or strong priors, such as constant brightness [19] or static scene [20], which restricts their practical applicability.

Besides to hand-crafted methods, deep-learning approaches can also be employed for TTC estimation. One alternative approach is to use scene flow estimation methods [28, 40, 35, 21, 45] that predict both depth and velocity simultaneously, enabling the generation of pixel-level TTC estimation maps. However, these methods depend on accurate optical flow information, which can result in significant computational overhead. Recently, [1] proposed Binary TTC that bypasses optical flow computation and directly computes TTC via estimating scale ratio between consecutive images. While these learning-based methods may produce more promising results, they require a larger amount of data with annotated scene flow ground-truth. Due to the expensive labeling cost, scene flow datasets are mostly obtained through synthesis, leading to a domain gap for real-world applications.

In addition to the aforementioned literature, single object tracking (SOT) [3, 23, 41, 9, 46] can also infer the scale ratio between the template and the tracked object, serving as an alternative approach to estimating TTC. However, these methods often rely on large downsampling rates, which may lead to the bounding box estimation not accurate enough to meet the requirements of TTC estimation. The baseline method which utilized SOT models in our experiment validates the similar effect.

**Datasets.** Contrary to previous TTC estimation studies, our proposed dataset facilitates the expansion of hand-craft features from solely relying on RGB images to utilizing the features extracted by neural networks. Moreover, we propose a method to address the problem of estimating the TTC by classifying the scale ratio. And we extend the implementation on RGB images to feature maps extracted using deep learning models, thereby significantly enhancing the accuracy of TTC estimation.

For TTC estimation, several datasets collected in real scenes are available. For example, [13] proposed a multi-person tracking dataset with stereo depth details, and

Table 1: Comparison with several autonomous vehicle (AV) datasets. "2D-to-3D" indicates the presence of tightly bounded 2D box annotations with corresponding 3D bounding boxes, which is crucial for TTC estimation.U and H in scenes indicate Urban and Highway respectively. And we report the average velocity of the recording platform in the training set for comparison.

|                   | KITTI     | NuScenes  | Waymo    | Ours           |
| ----------------- | --------- | --------- | -------- | -------------- |
| Scenes            | U         | U         | U        | **U+H**        |
| Frequency ($hz$)  | 10        | 2         | 10       | 10             |
| Range ($m$)       | [0,125]   | [-80,80]  | [-75,75] | **[-160,400]** |
| 2D-to-3D          | ✔         | ✘         | ✘        | ✔              |
| Avg Speed ($m/s$) | -         | 5.1       | 6.9      | 19.1           |
| Ann. Frames       | 15K       | 40K       | 200K     | **1M**         |
| Boxes             | 200K      | 1.4M      | **12M**  | 1M             |

[26] presented a large-scale dataset for TTC estimation in indoor scenes. However, these datasets mainly focus on low speed scenarios and are not suitable for direct application to TTC estimation in driving scenarios. In addition to datasets specifically designed for TTC estimation, some datasets have been synthesized for scene flow tasks and can provide detailed depth information, which are suitable for TTC estimation. For example, KITTI scene flow [28] proposed an outdoor scene flow dataset containing 400 dynamic scenes collected from KITTI [17]. These scenes are annotated using 3D CAD models for vehicles in motion and manually mask non-rigid moving objects. The Driving [27] dataset proposed a synthetic stereo video dataset rendering in realistic style. However, these datasets

are constrained by synthetic and limited scenes, which result in domain gaps with real scenes. Except to the scene flow datasets, some RGBD datasets [33, 34, 39], equipped with comprehensive depth annotations, can be utilized to train depth estimation models, which in turn can be used for TTC estimation. However, the majority depth annotations in these datasets are typically confined to a relatively small range (less than 50 meters) and the number of scenes available is limited. In addition to the aforementioned datasets, several large-scale datasets proposed for autonomous driving, such as [5, 7, 26, 38], offering comprehensive annotations like 3D LiDAR bounding boxes that can be used to generate TTC ground-truth. Nevertheless, these datasets were not specifically tailored for TTC estimation and presented issues like unbalanced TTC distribution. Furthermore, these datasets were mainly collected from urban areas, lacking data for highway scenarios where the estimation of TTC is particularly important. Please refer to Table 1 for a comparison of various datasets. Compared to the aforementioned datasets, our proposed dataset holds the distinct advantage of being large-scale and recorded in real scenarios, encompassing both urban and highway scenes.

## 3  Discussion

A common question related to TTC is that is TTC only applicable for low level assistant driving system? Why do we need TTC when we have distance and velocity measurement in advanced assistant or autonomous driving system? We argue that there are two main reasons for the essence of TTC: First, among all the three physical properties of one object (distance, velocity and time to contact), TTC is the only direct observation from monocular image. Monocular depth estimations are mostly from fully data-driven aspect, which highly relies on the recognition of the objects and scenes, which suffers from out of domain issue seriously [11]. For velocity, the situation is similar to that of depth estimation. However, TTC estimation does not require the recognition of semantic of objects (can even be achieved by pixel photometric loss), thus has better generalization in corner cases. Second, for a high level autonomous driving system, redundancy design is indispensable. Even though we have distance and velocity observation, TTC can also be fused into these observations in subsequent perception fusion modules, which serves as another independent safety guarantee.

## 4  TTC Dataset

### 4.1  Data Collection

The dataset consists of two parts, including a majority of data from real scenes and a minority of data from NeRF virtual rendering. After obtaining raw sensor data, we consider a sequence that contains a number of consecutive frames of a vehicle as a sample. To gather valuable data, we discard truncated objects within the camera plane and filter out 2D boxes smaller than $15 \times 15$ pixels.

**Real Scenes.**  For real-world scenarios, raw data was collected by our commercial trucks. We capture image data using three frontal and two backwards cameras with same $1024 \times 576$ resolution. We adopt 2D object detection and tracking algorithms on the images to obtain 2D bounding boxes and corresponding track ID for other vehicles. And the LiDAR and Radar data are adopted to generate accurate depth and velocity of them. The sampling rate is 10Hz. Detailed sensor specification can be found in the appendix. After applying these rules to filter the raw data, we observe that the resulting data distribution is highly imbalanced across different TTC ranges. For example, vehicles with small TTC are extremely rare in driving scenes, especially for vehicles lie in the same lanes as ego vehicle. However these are the cases which FCW and AEB should focus on. In such conditions, data collection without rebalancing may result in lack of these valuable scenarios. To overcome this challenge, we pre-define a data distribution and sample the data accordingly. The sampling weights for the TTC intervals, specifically (0,3], (3,6], (6,10], (10,15], and (15,20], is set to 14%. For the TTC range of [-20,0), the sampling weight is designated at 30%.

**NeRF Scenes.**  Despite thorough data collection efforts, we discover samples with small TTC values within the same lane are still extremely scarce, which is a crucial scenario in automated driving and

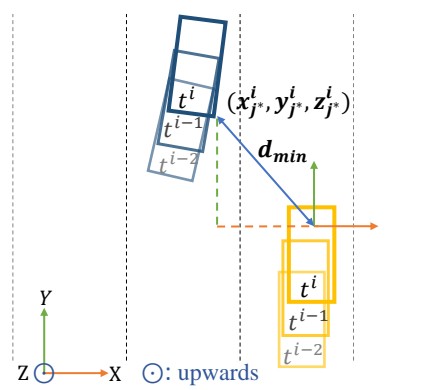

Figure 2: Relative position between ego vehicle and an object in bird-eye-view. Only three frames are plotted for brevity.

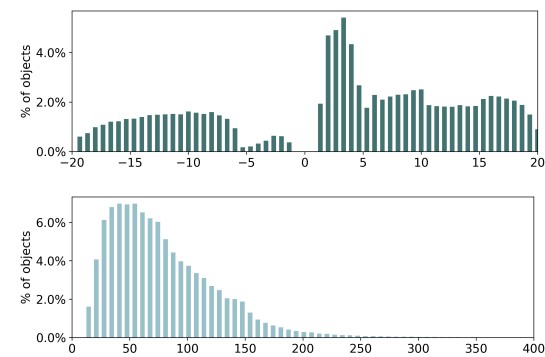

Figure 3: Histogram of TTC GT and relative depth of the training set.

ADAS. To supplement the absence of data in particular conditions, we adopt an internal undisclosed project which is developed based on Instant-NGP [31] to render novel scenes. Briefly speaking, the background models and object models are firstly extracted and trained separately. And then we can form a new scene and render them together. Given a specific object, we pre-define some scripts in which the TTC of the object is distributed between 0 and 6. The preset script can be found in our appendix. After obtaining NeRF rendered images, we organize them with the same format as real scenes data, which serves as an optional component within the training set.

## 4.2 Annotation

In each sequence, we provide the TTC ground-truth for every frame as the annotation. In the following, we will describe how we generate the TTC annotation. Given a frame, we first run 2D detection on the image and 3D detection on LiDAR. The long range LiDAR detection algorithm which reliably covers [-160, 400] meter range. Then, we could obtain its corresponding 2D detection box by projecting the 3D box to the image plane then picking the 2D detection box which has highest IoU between the projected box. In the vehicle coordinate system, one corner of the 3D bounding box could be denoted as $x_j, y_j, z_j$ where $j \in \{1, 2, ..., 8\}$ is the corner index. Here, we take the y-coordinate of the corner point which is the closest to ego as the depth of this object:

$$j^* = \underset{j \in \{1,2,...,8\}}{\arg\min} (\sqrt{(x_j^i - 0)^2 + (y_j^i - 0)^2 + (z_j^i - 0)^2}), \text{ where } y^i = y_{j*}^i, \qquad (2)$$

where $y^i$ is the depth of the vehicle in $i$-th frame. Here, we assume the relative velocity between vehicle and ego is constant in a short time interval (e.g. $q$ frames) to acquire a stable estimation of the velocity. Given the depth of the vehicle in the past $q$ frames, we fit the depth to obtain the relative velocity $v^i$ by RANSAC [15] algorithm. We set the value of $q$ to 10 by default. It is worth noting that the velocity is acquired prior to sequence splitting, thereby $q$ may be greater than the sequence length. After obtaining the depth and relative velocity of current frame, the TTC ground-truth could be obtained by $\tau^i = y^i/v^i$.

Since the camera planes are almost parallel to the XZ plane in the vehicle coordinate and the origins of these coordinate systems are highly aligned, we regard the TTC value calculated in the vehicle coordinate system as the TTC ground-truth for camera planes. The annotation process is illustrated in Fig 2. One may challenge that using past 10 frames to fit the velocity may result in latency in velocity estimation especially for sudden break. To remedy this issue, we first generate TTC ground-truth with different $q$ (e.g. 3, 5 and 10) and consider the vehicle with varied TTC values as an accelerating or decelerating object. For object with varied TTC values, we select the one closest to the TTC computed by the velocity of radar sensor as the ground truth. Note that we do not directly utilize radar sensors for all objects since they could not cover too distant objects. **The data annotations are then manually checked by our annotation team to ensure the quality**

## 4.3 Dataset Statistics

We construct the dataset following the pre-defined rules and annotation pipeline, resulting in 206K sequences comprising over 1M frames from real scenes, as well as 1K sequences from 6.0K NeRF rendered images. We split the sequences from the real scenes based on their recorded date to training, validation and test sets, yielding 149.1K, 28.8K, and 28.6K sequences respectively. For better understanding the data distribution, we plot the histogram of the TTC ground-truth and depth in the training set in Fig. 3. The distribution in validation and test set is similar. Note that the far away samples are rare because we set a minimum 2D bounding box size. More detailed information about the dataset statistics is provided in our appendix.

## 4.4 Task Definition

As the tracklet of a vehicle is collected in the format of fixed length sequences, we formulate the TTC estimation task in sequence level. For a sequence of a specific vehicle, we provide six consecutive frames and their corresponding 2D bounding boxes as input. The last frame in the sequence is considered as the target frame while the rest of the frames serve as references. With all frames and boxes available in the sequence, the objective is to predict the TTC value of the object in the target frame or equivalently relative scale ratio between the target box and the referenced one.

## 5 Metrics & Method

In this section, we first review the relationship between TTC estimation and scale ratio briefly. Then, we explicate the evaluation metrics for our TTC estimation task. Subsequently, we introduce our approach, which comprises two variants: the pixel MSE approach and its deep learning counterpart.

## 5.1 Estimate TTC via Scale Ratio

As shown in Sec. 4, TTC could be obtained by depth and its rate of change. However, estimating the depth and relative velocity of an object directly with only a monocular camera is very challenging. To address this issue, researchers proposed to estimate the TTC of frontal-parallel, planar non-deformable objects according to the change of object scales. As described in [19], we can obtain the image size of a frontal-parallel, non-deformable object of size $S$ at distance $y$ as:

$$s = fS/y, \tag{3}$$

where $f$ is the focal length of the camera. For an object without rotation, TTC can be formulated as a function of object size in image space by combining Eq. (1) and (3):

$$\tau = \frac{t_1 - t_0}{1 - \frac{s(t_0)}{s(t_1)}} = \frac{t_1 - t_0}{1 - \alpha}, \tag{4}$$

where $s(t_0)$ and $s(t_1)$ are the sizes of image of an object in frame $t_0$ and $t_1$ correspondingly. Thus, the estimation of TTC can be simplified as a scale ratio estimation problem which can be done with only observations in image space. With the development of deep learning, modern object detectors or tackers could produce relatively accurate 2D bounding boxes for vehicles. Given the detection or tracking bounding box in consecutive frames of a vehicle, one intuitive idea is that we can use the ratio of the box or mask area to accomplish the scale ratio estimation. However, are these bounding boxes accurate enough for the TTC estimation task and does there exist more accurate scale ratio estimation algorithms under such conditions? We try to answer these questions via experimental validation on the proposed dataset.

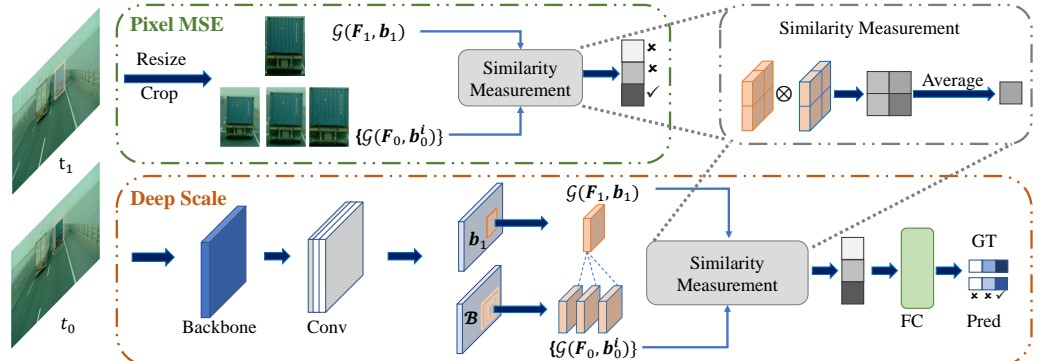

Figure 4: Framework of our proposed methods. After aligning the size between contents in $\mathbf{b}_1$ and $\mathcal{B}$, an operator $\bigotimes$ is applied to measure their similarity. For simplicity, we only illustrate three scale rates of $\mathcal{B}$. Some operations such as center shift are omitted for brevity. The green dashed box and the orange dashed box represent Pixel MSE and Deep Scale, respectively.

## 5.2 Evaluation Metrics

Before introducing the detailed design, we need to design the metrics for evaluation. Here, we adopt Motion-in-Depth (MiD) error and Relative TTC Error (RTE) as performance indicators, which could be denoted as:

$$\text{MiD} = ||\log(\alpha) - \log(\hat{\alpha})||_1 \times 10^4,$$
$$\text{RTE} = ||\frac{\tau - \hat{\tau}}{\tau}||_1 \times 100\%,$$
(5)

where $\alpha$ and $\hat{\alpha}$ mean the ground-truth and predicted scale ratios while the $\hat{\tau}$ denotes predicted TTC value. The scale ratio ground-truth $\alpha$ is obtained by Eq. (4). MiD is utilized in previous works [45, 1] to describe the TTC error indirectly from the perspective of scale ratio. Due to the instability of TTC at larger values, **we prioritize the MiD as the primary metric**. With the purpose of revealing more information from the evaluation metrics, we partition several TTC intervals, namely crucial(c) / small(s) / large(l) / negative(n)[1], which correspond to TTC values of 0~3, 3~6, 6~20, and -20~0, respectively. We mark them on the indices of the RTE and MiD. The threshold for crucial cases is determined by rounding the typical TTC threshold of 2.7 seconds used in FCW systems [36, 48]. This division allows a more detailed analysis of the performance for the TTC estimation algorithms in different TTC intervals, providing a better understanding of their limitations and strengths. During the prediction, TTC values that exceed the predefined range will be truncated to the boundary value.

## 5.3 Our Design

**Formulation.** We denote the 2D bounding box as $\mathbf{b} = [x, y, w, h]$ where $x, y$ represent the center coordinate and $w, h$ denote the width and height. Given a reference frame $\mathbf{F}_0$ and a target frame $\mathbf{F}_1$, $\mathbf{b}_0 = [x_0, y_0, w_0, h_0]$ and $\mathbf{b}_1 = [x_1, y_1, w_1, h_1]$ are bounding boxes of a specific object in these two frames. The core idea of our methods is to estimate the relative scale ratio of this vehicle between the reference frame and the target frame. A straightforward approach is simply adopting the scale ratio between $\mathbf{b}_0$ and $\mathbf{b}_1$ as the result. However, this strategy is largely influenced by the precision of detection algorithms. In our methods, we estimate the scale ratio change in these two frames in pixel space or feature space, yielding two kinds of implementation: Pixel MSE and Deep Scale. For the reference frame, we enumerate $n$ different scale ratios to obtain a series of scaled boxes $\mathcal{B} = [\mathbf{b}_0^{\alpha_1}, \mathbf{b}_0^{\alpha_2}, ..., \mathbf{b}_0^{\alpha_i}, ..., \mathbf{b}_0^{\alpha_n}]$ where $\mathbf{b}_0^{\alpha_i} = [x_0, y_0, \alpha_i w_1, \alpha_i h_1]$. Then, we crop $\mathbf{F}_0$ via $\mathcal{B}$ and resize them to a target size of $W, H$, which could be denoted as $\mathcal{G}(\mathbf{F}_0, \mathbf{b}_0^{\alpha_i})$ for the $i$-th scale ratio, where $\mathcal{G}(\mathbf{F}, \mathbf{b})$ denotes the crop $\mathbf{b}$ on $\mathbf{F}$ and resize it. Similarly, we could get $\mathcal{G}(\mathbf{F}_1, \mathbf{b}_1)$ for the target frame. Finally, we use an operator to measure the similarity between $\mathcal{G}(\mathbf{F}_0, \mathbf{b}_0^{\alpha_i})$ and

---
[1]Negative TTC indicates away from the observer.

$\mathcal{G}(\mathbf{F}_1, \mathbf{b}_1)$, yielding $n$ similarity scores. With five frames free to reference, taking different frames as the reference will produce different scale ratios. To address this issue, we convert scale ratio to corresponding TTC value via Eq. (4) and convert it to the scale ratio under the setting of 10Hz when computing MiD. We list the relationship between different scale ratios of same $\tau$ in the appendix.

**Pixel MSE.** We can measure the similarity between $\mathcal{G}(\mathbf{F}_0, \mathbf{b}_0^{\alpha_i})$ and $\mathcal{G}(\mathbf{F}_1, \mathbf{b}_1)$ in image space with Mean Squared Error (MSE). The weighted sum of the top $k$ similar scale ratios is adopted as the final estimation and the weight is normalized by the reciprocal of MSE. The top of Fig. 4 illustrates the pipeline of Pixel MSE.

**Deep Scale.** For the deep version, we first input two images into a backbone network for feature extraction. Afterwards, the grid sampling operation is used to align the features of different box sizes into one fixed size. Then, we calculate the similarity of each position in the two feature maps via cosine similarity, yielding a similarity map $\mathbf{S}_i$. Afterwards, the similarity score of scale $\alpha_i$ is obtained by adopting a Global Average Pooling (GAP) operation to $\mathbf{S}_i$. Then we concatenate the similarity scores of different scale ratios and use a Fully-Connected (FC) layer to obtain the final prediction. During training, binary cross-entropy (BCE) loss is used and we adopt the strategy proposed in [24, 47] to convert the scale ratio ground-truth to a size $n$ vector as soft label. Similar to Pixel MSE, we apply a top $k$ weighted sum operation to get the final results and the weight is defined as the sigmoid of the FC output. For fast inference, we adopt a convolutional layer followed by a stage of modified CSPNet [42] used in [16] as our backbone. After obtaining backbone features, we fed them into one transposed convolutional layer and two convolutional layers before similarity measurement. To capture subtle details, all the stride in the network is set to 1 except the first and the transposed convoultional layer which are set to 2 yielding a feature map with the same size of input. The framework is illustrated at the bottom of Fig. 4.

**Center Shift.** The boxes predicted by the object detection model may be inaccurate, which will bring misalignment between the centers of reference and target boxes. To address this problem, we introduce a center shift operation. Specifically, we enumerate an offset of $[-c, c]$ along height and width direction, respectively, which yields a total of $(2c+1) \times (2c+1)$ candidates. After obtaining $(2c+1) \times (2c+1)$ similarity scores for a single scale, we adopt the highest score as the final score for both Pixel MSE and Deep Scale. Our experiments show that this operation will bring significant improvement in terms of MiD and RTE with little time cost.

# 6 Experimental Validation

## 6.1 Implementation Details

**Pixel MSE.** For Pixel MSE, we validate its performance on validation, and test set. The target size $W, H$ after interpolation is set to the size of $\mathbf{b}_1$. The scale ratio is set to the range of $[0.65, 1.5]$ to cover samples with different scale ratios in the training set. The number of scale bins $n$ is 125, the top $k$ for the weighted sum and the $c$ for center shift are 3. Besides, the detection boxes are manually expanded with a maximum ratio of 1.1 (if the expanded boxes do not exceed the image boundary) to reduce the influence of inaccurate detection results.

**Deep Scale.** In terms of Deep Scale, we train it for 36 epochs on the train/train+val set dataset for evaluation on val/test respectively with a batch size of 16 using SGD [18]. The image is resized to $1024 \times 576$ for both training and test phases. We adopt random color on HSV space as data augmentation during training. The weight decay and SGD momentum parameters are set to 0.0005 and 0.9, respectively. We start from a learning rate of $10^{-4} \times$ BatchSize and adopt cosine learning rate schedule. The target size is set to $50 \times 50$ for grid sample as larger size does not bring more benefits. The input channel for the backbone is set to 12, while the channel for the three followed convolutional layers is set to 24. The kernel size of the convolutional layers is 7 while the transposed one is 3. For the scale range and box expansion, we keep them the same as Pixel MSE. Besides, the

number of scale bins $n$, the top $k$ for the weighted sum and the $c$ for center shift are set to 20, 4 and 1 respectively. We test the latency of all models with FP16 and batch size of 1 on a 3090 GPU.

Besides the aforementioned methods, we further propose two baselines termed as Detection and SOT in Table 4. For Detection, the scale ratio is obtained by simply computing the ratio between the area of the target box and the reference box. As for SOT, given a target box, we adopt a state-of-the-art (SOTA) SOT tracker [46] to obtain a reference box and then estimate the scale ratio as the same as in Detection. Additionally, we include results from an internal monocular depth algorithm and a LiDAR detection + tracking algorithm for a comprehensive evaluation. Details of these algorithms are provided in our

Table 2: Main results of different methods on the validation set. The † means the result is obtained under padding NeRF data. The % after RTE is omitted for brevity.

| Methods | MiD | $\text{MiD}_c$ | $\text{MiD}_s$ | $\text{MiD}_l$ | $\text{MiD}_n$ | RTE |
|---|---|---|---|---|---|---|
| Detection | 213.9 | 675.4 | 305.1 | 112.4 | 115.3 | 58.1 |
| SOT [46] | 200.8 | 641.1 | 261.1 | 77.4 | 158.8 | 57.1 |
| Pixel MSE | 41.0 | 57.4 | 36.5 | 32.5 | 48.4 | 29.9 |
| Depth | 62.3 | 111.9 | 74.6 | 36.1 | 68.4 | 47.3 |
| LiDAR | 6.4 | 12.5 | 6.0 | 5.3 | 3.3 | - |
| Deep Scale | 14.4 | 27.1 | 16.4 | 10.9 | 13.5 | 12.1 |
| Deep Scale† | 14.3 | 26.5 | 15.2 | 10.8 | 13.5 | 12.0 |

appendix. Although there are other methods available, such as [45, 1], the models released by the authors achieve poor results in our dataset due to the domain gap, and no training codes are provided.

## 6.2 Main Results

The detailed comparison between various methods is presented in Tab. 4. Due to page constraints, we report only their performance on the MiD metric and overall RTE. As observed, the RTE predicted by box detection or tracking methods is approximately 50%, which is inadequate for practical applications. In contrast, our Pixel MSE method demonstrates significantly lower MiD errors, indicating more accurate scale ratio estimations and consequently, lower RTEs. Among learning-based methods, our Deep Scale significantly outperforms the depth estimation method, although it remains inferior to the LiDAR detection + tracking algorithm. Nevertheless, it achieves the best performance among methods that utilize images. **More detailed comparison, ablations and visualizations could be found in the appendix.**

## 7 Limitations

Our work is a large-scale benchmark for Time-To-Contact estimation, but there are still some limitations that need to be addressed in future works. In regard to our dataset, the collected data primarily focuses on trucks and cars in highway and urban scenarios, lacking more diverse categories that are commonly found in autonomous driving datasets, such as cyclists and pedestrians. Besides, we have to acknowledge that due to the inherent differences in distribution between our dataset and real-world scenarios, there may be potential challenges when directly applying the model trained on our dataset to real-world contexts.

Furthermore, the baseline methods proposed in this study operate under the assumption that objects exhibit frontal-parallel characteristics and are non-deformable. However, it is essential to acknowledge that real-world conditions are considerably more intricate, and these methods may yield suboptimal performance when the underlying assumption is not met. Additionally, as we mentioned earlier, our methods are sensitive to the alignment between the box center of the target and reference frame, which poses another limitation.

## 8 Conclusion

In this work, we built a large-scale TTC dataset and provided a simple yet effective TTC estimation algorithm as baselines for the community. Our dataset is characterized by its focus on objects in driving scenes, which contains both urban and highway scenarios and covers a wider range of depth. We hope that our proposed dataset could facilitate the development of TTC estimation algorithms.

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

# Appendix

## Sensor Specification

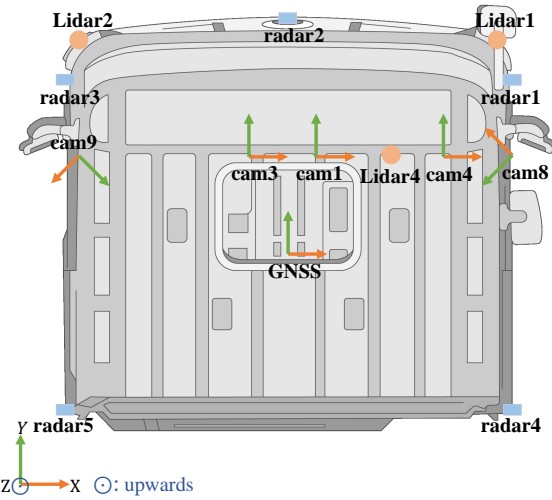

Figure 5: Sensor layout and coordinate system. The coordinate system of radars and Lidars are omitted for brevity. And GNSS means the Global Navigation Satellite System.

In our research, we conducted a comprehensive data collection process utilizing five cameras with varying focal lengths, five radar sensors, and three Lidar sensors. The spatial arrangement of these sensors is visually depicted in Fig. 5. Furthermore, we provide detailed specifications of the cameras in Tab. 3.

## More details about data statistic

Real scenes comprise 10.8% urban/suburban and 89.2% highway data. However, rare cases with small TTC on the same lane ([0,6]) make up only 0.02% of real scenes. To address this, we have supplemented these rare cases using data synthesized by NeRF. The synthesized data, constituting 5K sequences, represents highway scenes and is exclusively used for the training set. The data distribution of training set for object sizes and time-of-day are shown in Fig 6. As data in training, validation, and test sets are randomly split, their distributions are

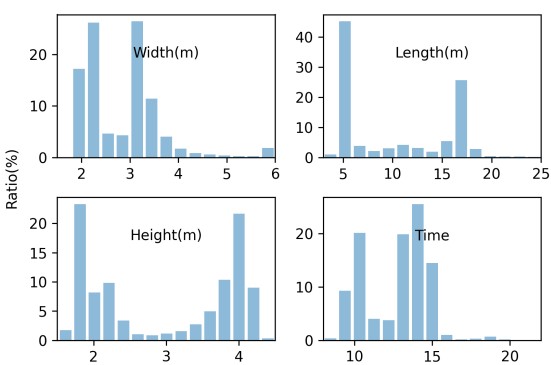

Figure 6: Distribution of object sizes and time-of-day.

Table 3: Camera specification. all images captured by the cameras are subjected to downsampling and cropping, resulting in a size of 1024x576 pixels. The camera's horizontal field of view (HFOV) refers to the angular range covered by the camera along the y-axis in the x-y plane of the camera sensor frame.

| Camera | 1 | 3 | 4 | 8 | 9 |
|--------|---|---|---|---|---|
| HFOV | ±63.2° | ±40.4° | ±18.4° | ±63.2° | ±63.2° |

consistent, so we've omitted the latter two. Besides, the geographical distribution of the dataset spans two cities: Shanghai and Hebei in China.

## More explanation on MiD computing

In the Sec 5.3 of our main paper, we first convert the predicted scale ratio under different FPS settings to corresponding TTC value and then convert the TTC value to the scale ratio under the setting of 10Hz. Actually, the scale ratios computed under different FPS settings could convert to each other. Since the TTC ground-truth for the target frame is specific, we could get the relationship between the scale ratios (*e.g.* $\alpha_m, \alpha_n$) under different FPS settings (*e.g.* $FPS_m, FPS_n$) by Eq.(5) in our paper:

$$\alpha_m = \frac{1}{\frac{FPS_n}{FPS_m}(1/\alpha_n - 1) + 1}. \tag{6}$$

In practice, the scale ratios obtained under different FPS settings are converted to the 10Hz setting via Eq (6) directly.

## Detailed Results

Table 4: Detailed results of different methods. The † means the result is obtained under padding NeRF data. The % after RTE is omitted.

| Methods | Validation Set | | | | | | | | | |
|---|---|---|---|---|---|---|---|---|---|---|
| | MiD | $\text{MiD}_c$ | $\text{MiD}_s$ | $\text{MiD}_l$ | $\text{MiD}_n$ | RTE | $\text{RTE}_c$ | $\text{RTE}_s$ | $\text{RTE}_l$ | $\text{RTE}_n$ |
| Detection | 213.9 | 675.4 | 305.1 | 112.4 | 115.3 | 54.2 | 58.1 | 56.3 | 55.0 | 50.2 |
| SOT [46] | 200.8 | 641.1 | 261.1 | 77.4 | 158.8 | 52.8 | 57.1 | 50.9 | 48.1 | 58.4 |
| Pixel MSE | 41.0 | 57.4 | 36.5 | 32.5 | 48.4 | 29.9 | 11.3 | 13.0 | 31.0 | 44.3 |
| Depth | 62.3 | 111.9 | 74.6 | 36.1 | 68.4 | 47.3 | - | - | - | - |
| LiDAR | 6.4 | 12.5 | 6.0 | 5.3 | 3.3 | - | - | - | - | - |
| Deep Scale | 14.4 | 27.1 | 16.4 | 10.9 | 13.5 | 12.1 | 6.3 | 8.7 | 13.0 | 14.8 |
| Deep Scale† | 14.3 | 26.5 | 15.2 | 10.8 | 13.5 | 12.0 | 6.2 | 8.4 | 12.9 | 14.6 |

We report detailed results of different methods on validation set in Tab. 4 of both MiD and RTE. For the Depth method, we first estimate the depth using a mono depth estimation model and then utilize RANSAC to fit the TTC value. Our experiments on the validation set demonstrate that the relative error of the depth estimation is only 10.7%. However, the relative TTC error and MiD error are significantly worse than our proposed method, reaching 47.3% and 62.3 respectively. The primary reason for such large errors is the inherent noise present in the depth estimation. For the LiDAR model, it is a internal sparse detector like SECOND [44] and FSDv2 [14]. The 3D bboxes in the training set were generated by an internal algorithm. The tracker we used is Simple Track [32]. The average BEV IoU between the 3D bboxes generated by internal algorithm and the manual annotations on the validation and test sets is 83.1%.

In addition to quantitative results, we also illustrate several cases in Fig. 7 for intuitive understanding. In the last column, we draw the scaled target box in the

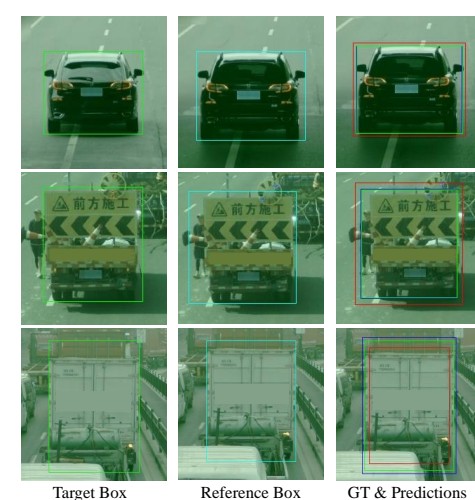

Target Box      Reference Box      GT & Predictions

Figure 7: Case study for our Pixel MSE and Deep Scale, best viewed in color. For the last column, box with green, red and blue color denote the scaled box obtained by GT, Pixel MSE and Deep Scale.

reference frame which is obtained by utilizing the center of the reference box and the scale ratio

Table 5: Influence of scale bins $n$ in Pixel MSE

| $n$ | 50 | 75 | 100 | **125** | 150 |
|---|---|---|---|---|---|
| MiD | 42.5 | 41.8 | 41.3 | 41.0 | 41.0 |
| RTE(%) | 31.9 | 31.9 | 30.8 | 29.9 | 30.0 |
| Time(ms) | 10.0 | 13.6 | 15.5 | 18.5 | 21.2 |

Table 6: Influence of scale bins $n$ in Deep Scale.

| $n$ | 10 | 15 | **20** | 25 | 30 |
|---|---|---|---|---|---|
| MiD | 16.8 | 14.9 | 14.4 | 14.4 | 14.4 |
| RTE(%) | 13.5 | 12.7 | 12.1 | 12.4 | 12.2 |
| Time(ms) | 11.7 | 11.9 | 12.0 | 12.2 | 12.4 |

Table 7: Ablation on the center shift operation for Pixel MSE and Deep Scale

| | Pixel MSE | | Deep Scale | |
|---|---|---|---|---|
| | **w/ S** | w/o S | **w/ S** | w/o S |
| MiD | 41.0 | 73.3 | 14.4 | 17.2 |
| RTE(%) | 29.9 | 37.6 | 12.1 | 14.6 |
| Time(ms) | 18.5 | 17.4 | 12.0 | 10.8 |

Table 8: Ablation on padding different amounts of NeRF rendered sequences.

| Seqs | MiD | $MiD_c$ | $MiD_s$ | $MiD_l$ | $MiD_n$ |
|---|---|---|---|---|---|
| 0 K | 14.4 | 27.1 | 15.5 | 10.9 | 13.5 |
| 3 K | 14.5 | 27.1 | 15.4 | 10.8 | 13.6 |
| **5 K** | 14.3 | 26.5 | 15.2 | 10.8 | 13.7 |
| 7 K | 14.6 | 26.7 | 15.4 | 11.2 | 13.8 |

obtained from various methods and ground truth. From the first and second cases, we can observe that the Deep Scale is more robust to the illumination changes and inaccurate detection boxes compared to Pixel MSE. In the last row, we showcase a failure case caused by a severely inaccurate detection box. As described before, our methods are sensitive to the alignment between the box center of the target and reference frame, which is also a limitation of our methods. These cases also reveal the key difference between TTC estimation and tracking task: *TTC estimation requires far more accurate estimation than tracking, while tracking usually focuses on complicated appearance change.* Furthermore, we have included visualizations in GIF format in the supplementary material located at `./Vis/monoDepth_visualization` to compare our proposed method and TTC estimation via depth estimation. These visualizations provide a more intuitive understanding of the noise in the depth estimation.

## Ablation Study

To validate the contribution of different components and hyper-parameters, we perform extensive experiments and report the results on the validation set unless otherwise specified. Default hyper-parameters are denoted in **bold**.

**Number of Scale Bin.** To probe the suitable scale bins for our Pixel MSE, we conduct ablation experiments, as shown in Table 5. The performance continues to boost until scale bin number reaches 125 and then becomes stable. Thus, we take the 125 bins as the default setting. To determine the optimal scale bin number $n$ for Deep Scale, we vary the value of $n$ from 10 to 30. As shown in Table 6, the MiD and RTE continuously decrease until $n = 20$, after which they tend to be stable. However, an increase in scale bins results in a larger computation cost. As a consequence, we set $n$ to 20 by default.

**Center Shift.** In this experiment, we present a comparison between the results obtained with and without center shift operation. We list the results for both pixel MSE and Deep scale, as shown in Table 7. The center shift operation is effective in reducing the estimation error.

**NeRF Augmentation.** To investigate the impact of supplementing NeRF data on the training process for the Deep Scale, we conducted ablation experiments by adding different numbers of NeRF sequences. The added NeRF sequences were randomly selected from the NeRF data we rendered. To avoid the potential impact of data randomness on the experimental results, we report the average results of five runs with different random seeds in Table 8. The results show that padding NeRF sequences can effectively reduce the MiD error in crucial scenes. However, too many NeRF sequences lead to a slight decrease in overall performance. Therefore, we use 5K rendered NeRF sequences in our experiments.

**On Target Size.** In this experiment, we ablate the target size $W, H$ for grid sample in Deep Scale and we keep $W = H$ when conducting ablation for convenience. Table 9 lists the result for different settings and we can observe that the performance continuously to boost until 50 and then becomes stable. As a consequence, we set the default target size to 50.

Table 9: Influence of the target size for grid sampling.

| Size | 10 | 25 | **50** | 75 | 100 |
|---|---|---|---|---|---|
| MiD | 20.0 | 14.9 | 14.4 | 14.8 | 14.9 |
| RTE(%) | 16.3 | 12.4 | 12.1 | 12.3 | 12.7 |

Table 10: Influence of the kernel size.

| K | 1 | 3 | 5 | **7** | 9 |
|---|---|---|---|---|---|
| MiD | 18.5 | 15.1 | 14.8 | 14.4 | 14.0 |
| RTE(%) | 15.2 | 12.3 | 12.3 | 12.1 | 11.8 |

**On Kernel Size.** Without large downsampling rate in our Deep Scale, the receptive field is mainly decided by the kernel size of the convolutional layers. To find the optimal kernel size, we train our model with the kernel size ranging from 1 to 9 and report the results in Table 10. Although increasing the kernel size can improve the performance, it comes at the cost of longer inference latency. As a trade off, we set the default kernel size to 7.

**On Plate Blur.** To verify whether the plate blur will influence the scale ratio estimation, we conduct ablation experiments. We test the MiD and RTE on the validation, and test set in w/ and w/o blur settings for the Pixel MSE. For the Deep Scale, we train the model on the blurred train/train+val set and report the result of val/test set in w/ and w/o blur settings. As shown in Table 11, for both methods, the plate blur operation brings negligible differences. As a conclusion, we take the dataset with blurred license plates as the release version.

**Number of Frame Gap.** In this experiment, we validate the influence of frame gap for both Pixel MSE and Deep Scale. The minimum and maximum scale ratio for different frame gaps are adjusted according to Eq. (6). For the Deep Scale, we train our model in different frame gaps. As for testing, we maintain the frame gap consistent with the training settings to ensure optimal results. We list detailed results in Table 12. Larger frame gap brings more obvious scale changes and thus benefits the classification process. As a result, we set the default frame gap to 5.

## More Visualization

In Fig. 8, we present more samples from our dataset, covering different ranges of TTC, meteorological fluctuations, and models rendered using NeRF. To get more intuitive understanding for different methods, we present more cases for visualization in Fig. 9. In the first column of Fig. 9, we plot the detection boxes of the target frames. In the second column, we show the boxes generated from detection and tracking model. For the last column, we show the scaled boxes obtained by GT, Pixel MSE and Deep Scale. As we can observe, the Pixel MSE produces unsatisfactory outcomes when object images encounter significant illumination changes or low quality, as exemplified in the first, second, and fourth cases. In contrast, the Deep Scale metric continues to perform robustly. Nonetheless, severe occlusion remains a challenge that adversely affects the performance of both Pixel MSE and Deep Scale, as demonstrated in the third case.

Additionally, we have included enhanced visualizations in GIF format within our supplementary material. These visualizations, which can be found in the `./Vis/prediction_visualization`

Table 11: Ablation on plate blur for our methods

| | | Pixel MSE | | Deep Scale | |
|---|---|---|---|---|---|
| | | **w/ blur** | w/o blur | **w/ blur** | w/o blur |
| Val | MiD | 41.0 | 41.0 | 14.4 | 14.4 |
| | RTE(%) | 29.9 | 30.0 | 12.1 | 12.1 |
| Test | MiD | 40.3 | 40.3 | 14.8 | 14.8 |
| | RTE(%) | 28.7 | 28.7 | 12.3 | 12.3 |

Table 12: Ablation on the frame gap for Pixel MSE and Deep Scale.

| | Gap | 1 | 2 | 3 | 4 | **5** |
|---|---|---|---|---|---|---|
| PixelMSE | MiD | 102.1 | 61.2 | 46.4 | 41.2 | 41.0 |
| | RTE(%) | 95.7 | 59.0 | 42.8 | 35.1 | 29.9 |
| DeepScale | MiD | 31.5 | 22.7 | 18.1 | 15.8 | 14.4 |
| | RTE(%) | 29.5 | 20.1 | 15.9 | 13.5 | 12.1 |

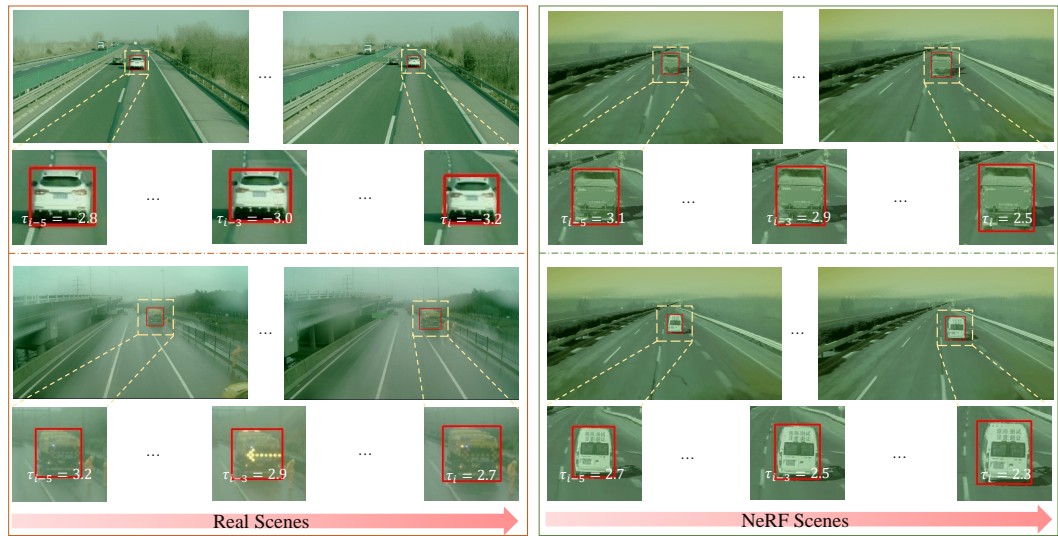

Figure 8: More visualization for samples in real and NeRF scenes.

and `./Vis/scene_visualization` directories, separately showcase qualitative results and various scene representations.

# NeRF Script

For the scripts used for NeRF rendering, we list them in Table 13. $v_{ego}$ and $v_{vehicle}$ denote the initial speed of ego and the target vehicle respectively. The $y$ denotes the relative distance in depth direction and we only consider the straight lane when setting the scripts. We permute and combine the speed of ego and vehicle to get more scenarios. For the rendered images, we also adopt the detection model to generate 2D bounding boxes and the truncated objects will be discarded to ensure the completeness.

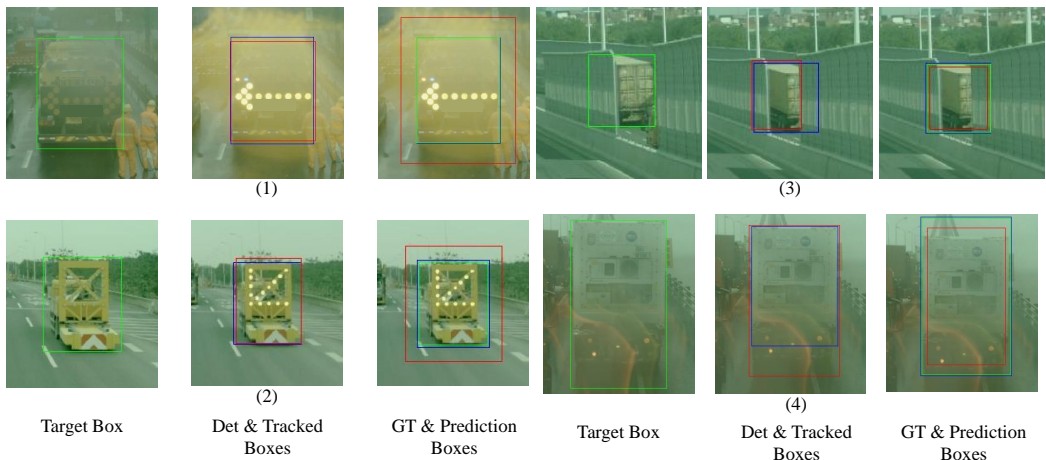

| Target Box | Det & Tracked Boxes | GT & Prediction Boxes | Target Box | Det & Tracked Boxes | GT & Prediction Boxes |

Figure 9: More visual comparison of the detection, tracking, and proposed methods, best viewed in color. In the second column, we use the blue and red color to distinguish the box from detection and tracking. For the last column, box with green, red and blue color denote the scaled box obtained by GT, Pixel MSE and Deep Scale.

Table 13: Script for NeRF rendering.

| No. | Initial Status | | | Script |
| --- | --- | --- | --- | --- |
| | $v_{ego}(km/h)$ | $v_{vehicle}(km/h)$ | $y(m)$ | |
| 1 | 40
60
80 | $v_{ego}$ | 50 | 1. Ego drives at $v_{ego}$ while the target vehicle decelerates at a speed of -3$m/s^2$.
2. After 3 seconds, ego decelerates with an acceleration of -4$m/s^2$ until its velocity matches that of the target vehicle. |
| 2 | 40
60
80 | $v_{ego}-20$ | 65 | 1. Ego drives at $v_{ego}$.
2. At a distance range of $(10, 50, 5)m$ to the target vehicle, ego performs a lane change at a constant lateral relative speed of $2m/s$, until it is completely in a different lane from the target vehicle. |
| 3 | 60
80 | 20
40 | 65 | 1. Ego gradually accelerates towards the target vehicle with an acceleration of range(0.5, 3, 0.5)$m/s^2$.
2. When the distance between ego and the target vehicle is within the range of $(10, 50, 5)m$, the target vehicle changes lanes with a constant lateral relative speed of $2m/s$ from a adjacent lane, until ego and the target vehicle are completely in same lane. At the same time, ego decelerates at -4$m/s^2$ within the same distance range of $(10, 50, 5)m$. |
| 4 | 60
80 | 60 | 65 | 1. Ego drives at $v_{ego}$ while the target vehicle decelerates at a speed of -3$m/s^2$.
2. After 3 seconds, ego decelerates with an acceleration of -4$m/s^2$ until its velocity matches that of the target vehicle. |
| 5 | 40
60 | 20
30 | 65 | 1. Ego drives at $v_{ego}$ while the target vehicle gradually accelerates with an acceleration range of (0.5, 3, 0.5)$m/s^2$.
2. At a distance range of $(20, 60, 4)m$ to the target vehicle, ego smoothly changes lanes with a lateral velocity range of $(0.5, 1.5, 0.2)m/s$, until ego and the target vehicle are completely in different lanes or the distance between them is less than $5m$. |
| 6 | 40
60
80 | $v_{ego}-30$ | 65 | 1. Ego drives at $v_{ego}$ while the target vehicle gradually accelerates with an acceleration range of (0.5,3,0.5)$m/s^2$.
2. When the distance to the target vehicle is in the range of $(10, 50, 4)m$, ego gradually decelerates with an acceleration of -(1, 4, 0.5)$m/s^2$ until ego's velocity matches the target vehicle's velocity or the distance between them is less than $5m$. |

