# OpenReview forum: "TSTTC: A Large-Scale Dataset for Time-to-Contact Estimation in Driving Scenarios"
_NeurIPS.cc/2024/Datasets_and_Benchmarks_Track — Submitted to NeurIPS 2024 Track Datasets and Benchmarks_

### Official Review · Reviewer_2cBV · 2024-07-20
**Review of Submission394**

**Rating:** 7
**Confidence:** 4
**Correctness:** The dataset if constructed in a sound…
**Clarity:** The paper is well-written and clear.

**Review:**

**Quality**

The dataset quality is high. The dataset API, document, and instructions are included, making the dataset ready for use.

**Clarity**

The paper is generally well-written and clear. The details of the dataset, metrics, and proposed baselines are presented. However, the entire structure needs to be improved to increase the continuity.

**Originality**

I think this dataset is original since previous datasets do not focus on the TTC estimation. The comparison between the provided dataset and previous datasets (KITTI, NuScenes, Waymo) is clear. Using Nerf to render novel scenes is also novel to supplement the absence of data in particular conditions.

**Significance**

The motivation is clear. Both AEB and FCW rely on the estimation of TTC. Vision-based methods are attractive due to their low cost. The dataset could have important value to the ADAS systems with learning-based models.

**Strengths:**

The problem of TTC estimation is important but has not been benchmarked before. So, this paper has great originality and significance. The dataset API, document, and instructions are included, making the dataset ready for use.

**Additional Feedback:**

N/A

**Documentation:**

Yes.

**Ethics:**

No.

**Limitations:**

Two main limitations are mentioned in the paper. The first is that the collected data primarily focuses on trucks and cars in highway and urban scenarios, lacking more diverse categories that are commonly found in autonomous driving datasets, such as cyclists and pedestrians. The second is the assumption that objects exhibit frontal-parallel characteristics and are non-deformable.

I think both limitations are acceptable considering the limited existing benchmark in this area.

**Opportunities For Improvement:**

1.	The Nerf scenes come from an internal undisclosed project based on Instant-NGP. Considering that Instant-NGP is an open-sourced tool, I suggest the authors also include the Nerf scene pipeline. Even a simplified version could be helpful for the community to generate more diverse scenes, maybe beyond the scope of TTC estimation.
2.	Table 2 is not mentioned in the context.
3.	Details about the experiment results and analysis are put in the appendix due to space limitations. I think these details are important. Maybe the authors can combine several sections to save some room. For example, the Section 3 discussion seems too short to be separated alone. Section 7 can be revised to be more precise and shorter, then combined with Section 8.

**Relation To Prior Work:**

A thorough related work section is included in the paper.

**Summary And Contributions:**

Time-to-Contact is a simple yet widely used metric for assessing collision risk. However, there is no large-scale TTC dataset for learning-based methods in real-world scenarios. In this paper, the authors present an object-oriented TTC dataset in the driving scene. They also generate clips using the latest Neural rendering sensor simulation, which is a novel idea for generating long-tail cases to augment the quantity of small TTC cases. They provide several simple yet effective TTC estimation baselines and evaluate them extensively on the proposed dataset to demonstrate their effectiveness.

---

> ### Author Rebuttal · Authors · 2024-08-16
>
> Thank you for your insightful comments and the time you have dedicated to reviewing our manuscript. We address your concerns in a few points as described below:
>
>
> ### Q1: Details of the Nerf scene pipeline.
> Thanks for this suggestion. The overview of the NeRF scene pipeline is complemented as follows:
>
> 1. **Object and scene segmentation**: We apply object detection and segmentation methods to the collected data.
> 2. **Rendering**: Both object and scene rendering are based on Instant-NGP, but with independent pipelines and models.
> 3. **Acceleration and database creation**: We use LiDAR point clouds to accelerate the rendering process, based on [1]. This allows us to construct an object bank and a scene library.
> 4. **Object-scene integration**: When rendering, we select a scene and choose objects from our bank that best match the scripted object trajectories. As the object positions are already known, their pixels are rendered using object NeRF, while the rest use scene NeRF.
> 5. **Shadow rendering**: We calculate shadow areas based on preset sun angles and 3D object bounding boxes, then render these areas using Blender.
>
>
> ### Q2: Table 2 is not mentioned in the context.
> We sincerely apologize for this oversight and appreciate the reviewer for pointing it out. We will correct this in the revised paper and carefully review our manuscript to avoid similar errors.
>
> ### Q3:  Rewrite Section 3 and Section 7 to include more experiment results.
> Thank you for your valuable feedback. We will merge Section 3 into Section 4 and make Section 7 more precise and concise, then integrate it into Section 8. The space saved will be used to add experimental results and analysis.
>
> ### Q4: Limitated scene and strong assumption.
> We deeply appreciate your understanding regarding the limitations of our work. We respectfully agree with these limitations and fully acknowledge that the current version of TSTTC falls short in object categories and scenarios. In subsequent works, we will include more common object categories that are typically found in widely-used autonomous driving datasets like Waymo and nuScenes. Additionally, when updating the dataset, we will focus on enhancing data diversity to ensure our dataset covers a broader range of regions, scenarios, weather conditions, and times of day. Besides, we will also provide more detailed annotations for objects in the revised version, including the detailed 3D annotation for objects, to facilitate future algorithm optimizations.
>
> We hope that these revisions could address your concerns. We thank you once again for your constructive feedback, which has significantly contributed to the improvement of our work.
>
> Reference:
>
> [1] Cao, Junyi, et al. "Lightning NeRF: Efficient Hybrid Scene Representation for Autonomous Driving." arXiv preprint arXiv:2403.05907 (2024).

---

### Official Review · Reviewer_CESF · 2024-07-24
**Review for TSTTC**

**Rating:** 6
**Confidence:** 3
**Clarity:** The paper is well-written and easy to…

**Review:**

The paper presents a comprehensive dataset with over 200,000 sequences to enhance TTC estimation using monocular cameras. It ensures a balanced distribution of TTC values by including NeRF-generated data and provides 2D/3D bounding boxes and TTC ground-truth. The authors propose two simple TTC estimation algorithms as benchmarks. Despite its value, the paper raises questions about the unique significance of focusing on TTC over 3D object detection and its necessity in autonomous driving. Additionally, the dataset's highway focus limits its environmental diversity.

**Strengths:**

- The paper introduces a comprehensive TTC dataset with over 200,000 sequences from real-world driving scenarios, enhancing the resources available for TTC estimation research.
- The authors ensured a balanced distribution of TTC values across the dataset by selectively sampling driving data and augmenting it with NeRF-generated data, addressing the scarcity of small TTC cases.
- The paper proposes two simple TTC estimation algorithms, providing benchmark methods for future research.

**Additional Feedback:**

The paper proposed a large-scale monocular TTC dataset including over 200K sequences covering a depth range of 400 meters. Though the technical novelty is somewhat weak, the proposed TTC estimation algorithms are helpful to serve as baseline methods for future study. Given that 3D detection appears to naturally encompass TTC and is potentially more versatile, what is the specific significance and unique value of focusing solely on TTC estimation? Is TTC estimation really necessary in autonomous driving system? It would be better if authors can clarify these problems clearly.

**Correctness:**

The authors provide a well-established dataset with adequate experiments, including comparison, ablations etc. And the experiments on TTC estimation algorithms proposed by authors perform well.

**Documentation:**

Clear documentation. And the dataset and code for TTC estimation algorithms proposed is accessible to the public with documentation.

**Limitations:**

- The dataset, primarily collected in highway scenarios, may be a bit limited and lacks diversity for other environments, compared with nuScenes and Waymo, which are diverse in terms of geographical locations, weather conditions, and types of urban and suburban environments.
- The paper mentioned that some RGBD datasets can also be used for TTC estimation, such as Diode, what are the absolute advantages of your proposed dataset compared to these existing datasets?
- Concerning method, the paper also mentioned that the baseline methods proposed in this study operate under the assumption that objects exhibit frontal-parallel characteristics and are non-deformable, while in real world, the underlying assumption is not met.



Ashutosh Saxena, Min Sun, and Andrew Y Ng. Make3d: Depth perception from a single still image. In AAAI, 2008.

Thomas Schops, Johannes L Schonberger, Silvano Galliani, Torsten Sattler, Konrad Schindler, Marc Pollefeys, and Andreas Geiger. A multi-view stereo benchmark with high-resolution images and multi-camera videos. In CVPR, 2017.

Igor Vasiljevic, Nick Kolkin, Shanyi Zhang, Ruotian Luo, Haochen Wang, Falcon Z Dai, Andrea F Daniele, Mohammadreza Mostajabi, Steven Basart, Matthew R Walter, et al. Diode: A dense indoor and outdoor depth dataset. arXiv preprint arXiv:1908.00463, 2019.

Wang, Z., Wei, Y., Rao, Y., Zhou, J., & Lu, J. 3D Point-Voxel Correlation Fields for Scene Flow Estimation. In TPAMI, 2023.

Holger Caesar, Varun Bankiti, Alex H Lang, Sourabh Vora, Venice Erin Liong, Qiang Xu, Anush Krishnan, Yu Pan, Giancarlo Baldan, and Oscar Beijbom. nuScenes: A multimodal dataset for autonomous driving. In CVPR, 2020.

Pei Sun, Henrik Kretzschmar, Xerxes Dotiwalla, Aurelien Chouard, Vijaysai Patnaik, Paul Tsui, James Guo, Yin Zhou, Yuning Chai, Benjamin Caine, et al. Scalability in perception for autonomous driving: Waymo open dataset. In CVPR, 2020.

**Opportunities For Improvement:**

- Develop and incorporate domain adaptation techniques to bridge the gap between synthetic NeRF-generated data and real-world data.  This can help improve the generalization capability of models trained on the dataset.
- Collect additional data from a wider variety of driving environments, including rural areas, off-road conditions etc.
- After the annotation team has manually completed the dataset annotations, is there a quality check process in place? If not, implementing one could be a valuable improvement.

**Relation To Prior Work:**

Clearly discussed.

**Summary And Contributions:**

The paper presents a new dataset specifically designed to facilitate Time-to-Contact (TTC) estimation using monocular cameras in driving scenarios. The authors gathered over 200,000 sequences from thousands of hours of driving data,  and for scenarios with small TTC values, they particularly employed NeRF to generate additional data, ensuring a balanced distribution of TTC values. The dataset includes 2D and 3D bounding boxes and TTC ground-truth for objects in each frame. Additionally, the paper proposes two TTC estimation algorithms and test them on the dataset.

The authors' contributions are as follows:

They present a balanced distribution of TTC values across the dataset by selectively sampling driving data and augmenting it with NeRF-generated data, and provide a detailed data generation process.They proposes two TTC estimation algorithms to show the research potential their proposed benchmark can open up.

---

> ### Author Rebuttal · Authors · 2024-08-16
>
> We deeply appreciate your insightful comments. Below, we address your concerns point by point:
>
>
> ### Q1: The proposed dataset lacks diversity.
>
> This version of the dataset mainly focuses on high-way cases, thus the categories such as pedestrians and cyclists barely appear under the circumstances. But, we totally agree with the reviewer that the elements in the dataset should be more diverse. In the future, we will iteratively include more kinds of objects in general  autonomous driving scenarios like the Waymo and nuScenes datasets. Besides, we will also  enhance data diversity in a broader range of regions, scenarios, weather conditions, and times of day.
>
> ### Q2: The advantages of the proposed dataset compared to RGBD datasets.
> Thank you for raising this concern. The advantages of the TSTTC dataset compared to RGBD datasets lie in two main aspects:
>
> Firstly, TSTTC is specifically constructed for driving scenarios, whereas RGBD datasets generally encompass a wide spectrum of scenarios. This difference results in a limited amount of driving scenario data within RGBD datasets. Additionally, the depth distribution of objects in RGBD datasets differs from that in the proposed dataset. For example, in the Diode dataset, the maximum depth is only 100 meters, with most depth annotations constrained to 50 meters. In contrast, TSTTC can cover objects up to 400 meters away, making it more suitable for TTC estimation in driving scenarios, especially for trucks and highway scenarios where a large perception range is crucial.
>
> Secondly, RGBD datasets typically lack temporal continuity between images. As a result, using RGBD data for TTC estimation reverts to the category of depth estimation, which inherently contains noise. We trained a monocular depth estimation model with our internal data and used it to estimate depth and fit the TTC value. Although the relative error of the depth estimation model was only 10.7% on the validation set, the relative TTC error and MiD error reached 47.3% and 62.3, respectively, which were significantly worse than our proposed method. In contrast, the TSTTC dataset includes over 200K object sequences, allowing the use of these continuous temporal sequences to supervise the TTC estimation algorithm, resulting in more accurate and smoother outcomes.
>
>
> ### Q3: The proposed method relies on  frontal-parallel and  non-deformable assumption.
> We thank you for bringing up this concern again. To address this issue, we will provide more detailed annotations for objects in the revised version, including the detailed 3D annotation for objects, to facilitate future algorithm optimizations.
>
> ### Q4: Is TTC estimation really necessary in autonomous driving system?
>
> Thank you for your insightful question. The unique value of focusing on TTC estimation in autonomous driving systems could be summarized as follows:
>
> (1). **Challenges in Pure Vision-Based 3D Detection**: Estimating the depth of distant objects using pure vision is inherently challenging, often resulting in imprecise object localization. Consequently, vision-based 3D object detection may not be able to provide accurate position information for distant objects. For NPC usage downstream , inaccurate position data  highly likely leads to erroneous object speed and TTC estimates, making it difficult to generate accurate and reasonable planning trajectories. In contrast, vision-based TTC estimation can yield relatively accurate results. As stated in the second question , the results from the depth estimation were significantly worse than our proposed method. We provide a more detailed comparison in *Table 2* of our main paper and visualization evidence in the *./Vis/monoDepth_visualization/* of the supplementary material. It will be deeply appreciated if you could check it.
>
> (2). **Robustness in Autonomous Driving Systems**: In a complex autonomous driving system, it is crucial to avoid reliance on a single algorithm to prevent system failure. A robust system often requires the fusion of multiple observations with different failure modes.  These failure modes represent the various ways in which algorithms or components of the system might malfunction or produce inaccurate results. Even though current 3D detection algorithms can provide relatively accurate 3D object detection results (e.g., when using LiDAR), there are still scenarios where performance may degrade, such as due to LiDAR noise. Therefore, an additional vision-based TTC estimation can help mitigate these issues and enhance overall system robustness.
>
> We will add explanations in the revised paper for clarity.
>
> Besides these concerns, we greatly appreciate the reviewer for pointing out the  opportunities for improvement. We will try to adopt the domain adaptation techniques to bridge the gap between generated NeRF data and real-world data.  And there is a double quality check in  the dataset annotation pipeline by our annotation team to ensure the quality of the annotation.
>
> We hope that these revisions adequately address your comments. We thank you once again for your constructive feedback, which has significantly contributed to the improvement of our work.

---

> > ### Comment · Reviewer_CESF · 2024-08-31
> >
> > Thanks for the rebuttal of the paper, that will clarify the paper to a more general audience. Please consider including the extra content in the main paper.

---

### Official Review · Reviewer_Su94 · 2024-07-25
**TSTTC: A Large-Scale Dataset for Time-to-Contact Estimation in Driving Scenarios**

**Rating:** 6
**Confidence:** 3
**Correctness:** Yes
**Clarity:** yes

**Review:**

Pros:
* This work introduces a substantial and diverse real-world dataset addressing a critical challenge in autonomous driving: Time-to-Contact estimation. The dataset's expansive scale, encompassing over 200,000 sequences, provides researchers with a robust foundation for developing and testing TTC estimation algorithms. Notably, the inclusion of both urban and highway scenarios enhances the dataset's versatility, enabling the development of more adaptable and generalizable autonomous driving systems that can navigate varied environments with improved safety and efficiency.
* The paper introduces an innovative approach to data augmentation by leveraging Neural Radiance Fields (NeRF) to generate synthetic data for rare scenarios. This novel application of NeRF technology allows the researchers to fill critical gaps in the dataset, particularly for uncommon but potentially dangerous situations like sudden braking events
* I enjoy reading section 3 which discuss the motivation behind building a TTC dataset as it's not commonly seen in autonomous vehicle community.



Cons:
* While the dataset provides a wealth of information on cars and trucks, it falls short in representing the full spectrum of road users encountered in real-world driving scenarios. Future iterations of the dataset could benefit from incorporating a wider range of object categories, such as pedestrians, cyclists, and various types of vehicle. In my understanding, vulnerable road user (VRU) is more important in AEB system.
* The accuracy of Time-to-Contact (TTC) estimation may be compromised in scenarios where the target vehicle and the ego vehicle are traversing a slope, introducing significant changes in elevation. This limitation is particularly concerning given that sloped terrain represents a critical scenario for Automated Emergency Braking (AEB) systems, where precise TTC estimation is crucial for effective collision avoidance and overall vehicle safety
* details of the NeRF algorithms is not discussed in this paper. The generated images with very small TTC are not evaluated and shown in this study.

**Strengths:**

listed in the Review section above

**Additional Feedback:**

None

**Documentation:**

Yes.

**Limitations:**

Yes.

**Opportunities For Improvement:**

Even though I appreciate the authors' effort of explain the motivation behind building a TTC dataset, I think it's questionable whether it would make enough impact. For example, having a independent vision-only 3D object detection module could reuse the common 3D object detection ground truth and serve as another independent safety guarantee.

**Relation To Prior Work:**

Yes

**Summary And Contributions:**

This paper presents TSTTC, a large-scale dataset for Time-to-Contact (TTC) estimation in driving scenarios. The dataset contains over 200K sequences from real-world driving data and synthetically generated scenes using neural rendering methods. The authors also propose two simple yet effective baseline methods for TTC estimation based on scale ratio estimation. Through extensive experiments, they demonstrate the effectiveness of their proposed dataset and baseline methods for advancing research on vision-based TTC estimation for autonomous driving applications.

---

> ### Author Rebuttal · Authors · 2024-08-16
>
> We sincerely appreciate your insightful comments. The following offers the responses to these comments.
>
>
> ### Q1: The proposed dataset lacks diversity.
> We emphasize that this version of the dataset mainly focuses on high-way cases, thus the categories such as pedestrians and cyclists barely appear under the circumstances. But, we totally agree with the reviewer that the elements in the dataset should be more diverse. In the future, we will iteratively include more kinds of objects in general  autonomous driving scenarios like the Waymo and nuScenes datasets. Besides, we will also  enhance data diversity in a broader range of regions, scenarios, weather conditions, and times of day.
>
> ### Q2:The proposed TTC estimation algorithm falls short in some scenarios.
> Thanks for pointing out this issue. In practical usage within our internal environment, we have observed that the proposed Deep Scale algorithm encounters failure cases when vehicles traverse slopes. This issue can be attributed to the assumption that objects are rigid and exhibit frontal-parallel characteristics, which does NOT always hold. We will add further clarification in the limitation section of our revision. Additionally, we will provide more detailed annotations for objects in the updated version, such as the detailed 3D annotation, to facilitate future algorithm optimizations.
>
> ### Q3: Details and evaluation of the NeRF algorithms.
> Thanks for this suggestion. The overview of the NeRF scene pipeline is as follows:
>
> 1. **Object and scene segmentation**: We apply object detection and segmentation methods to the collected data.
> 2. **Rendering**: Both object and scene rendering are based on Instant-NGP, but with independent pipelines and models.
> 3. **Acceleration and database creation**: We use LiDAR point clouds to accelerate the rendering process, based on [1]. This allows us to construct an object bank and a scene library.
> 4. **Object-scene integration**: When rendering, we select a scene and choose objects from our bank that best match the scripted object trajectories. As the object positions are already known, their pixels are rendered using object NeRF, while the rest use scene NeRF.
> 5. **Shadow rendering**: We calculate shadow areas based on preset sun angles and 3D object bounding boxes, then render these areas using Blender.
>
> To assess the quality of these generated images, we calculated the Fréchet Inception Distance (FID) between the generated NeRF images and the images in the validation set. The corresponding FID score is 12.1, indicating a good level of similarity between the generated and real images. Furthermore, we have complemented the visualization for generated samples with very small TTC values in the uploaded PDF. Samples with extremely small TTC values (<0.8) may not be fully displayed within a single image frame due to their proximity to the camera. We sincerely appreciate your attention to detail, which has helped us address this oversight in our study.
>
> ### Q4: The impact of the proposed dataset.
> We thank the reviewer for bringing out this reasonable concern. The unique value of focusing on vision-based TTC estimation in autonomous driving systems could be summarized as follows:
>
> (1). **Challenges in Pure Vision-Based 3D Detection**: Estimating the depth of distant objects using pure vision is inherently challenging, often resulting in imprecise object localization. Consequently, vision-based 3D object detection may be able to provide accurate position information for distant objects. For NPC usage downstream, inaccurate position data highly likely leads to erroneous object speed and TTC estimates, making it difficult to generate accurate and reasonable planning trajectories. In contrast, vision-based TTC estimation can yield relatively accurate results. Concretely,  we trained a monocular depth estimation model with our internal data and used it to estimate depth and fit the TTC value. Although the relative error of the depth estimation model was only 10.7% on the validation set, the relative TTC error and MiD error reached 47.3% and 62.3, respectively, which were significantly worse than our proposed method. We provide a more detailed comparison in *Table 2 of our main paper* and visualization evidence in the *./Vis/monoDepth_visualization/* of the supplementary material. It will be deeply appreciated  if you could check it.
>
> (2). **Robustness in Autonomous Driving Systems**: In a complex autonomous driving system, it is crucial to avoid reliance on a single algorithm to prevent system failure. A robust system often requires the fusion of multiple observations with different failure modes.  These failure modes represent the various ways in which algorithms or components of the system might malfunction or produce inaccurate results. Even though current 3D detection algorithms can provide relatively accurate 3D object detection results (e.g., when using LiDAR), there are still scenarios where performance may degrade, such as due to LiDAR noise. Therefore, an additional vision-based TTC estimation can help mitigate these issues and enhance overall system robustness.
>
> We will add explanations in the revised paper for clarity.
>
> We hope that these revisions could satisfactorily address your comments and thank you once again for your constructive feedback.
>
> Reference:
>
> [1] Cao, Junyi, et al. "Lightning NeRF: Efficient Hybrid Scene Representation for Autonomous Driving." arXiv preprint arXiv:2403.05907 (2024).

---

### Author Rebuttal · Authors · 2024-08-16

Dear reviewers and AC:

We would like to thank you for the effort you have dedicated to reviewing this manuscript, and the constructive comments that can further improve the quality of our work. We provide point-by-point responses to the concerns raised by the reviewers,

Thanks again.

Best regards,

The authors

---

### Decision · Program_Chairs · 2024-09-26

**Decision:**

Reject

**Comment:**

The paper proposes a large-scale and high-quality Time-to-Contact estimation dataset which are agreed by all reviewers is a novel, good and useful datasets for autonomous driving research. Reviewers all agree on the significance of the dataset contributions, although some key limitations exist, such as the lack of scene diversity (mostly high-way or urban) and the assumption that objects exhibit frontal-parallel characteristics and are non-deformable.